# Systematic Analysis of Cotton RING E3 Ubiquitin Ligase Genes Reveals Their Potential Involvement in Salt Stress Tolerance

**DOI:** 10.3390/ijms26010359

**Published:** 2025-01-03

**Authors:** Hao Li, Yizhen Chen, Mingchuan Fu, Liguo Wang, Renzhong Liu, Zhanji Liu

**Affiliations:** Key Laboratory of Cotton Breeding and Cultivation in Huang-Huai-Hai Plain, Ministry of Agriculture and Rural Affairs, Institute of Industrial Crops Shandong Academy of Agricultural Sciences, Jinan 250100, China; leehaul@hotmail.com (H.L.); chenyizhen0@126.com (Y.C.); fumingchuan@shandong.cn (M.F.); dale-wang@163.com (L.W.); liurenzh@163.com (R.L.)

**Keywords:** cotton, RING E3 ligase, gene duplication, expression analysis, salt stress

## Abstract

The Really Interesting New Gene (RING) E3 ubiquitin ligases represent the largest class of E3 ubiquitin ligases involved in protein degradation and play a pivotal role in plant growth, development, and environmental responses. Despite extensive studies in numerous plant species, the functions of RING E3 ligases in cotton remain largely unknown. In this study, we performed systematic identification, characterization, and expression analysis of *RING* genes in cotton. A total of 514, 509, and 914 *RING* genes were identified in *Gossypium arboretum*, *G. raimondii*, and *G. hirsutum*, respectively. Duplication analysis indicates that segmental duplication may be the primary mechanism responsible for the expansion of the cotton *RING* gene family. Moreover, the Ka/Ks analysis suggests that these duplicated genes have undergone purifying selection throughout the evolutionary history of cotton. Notably, 393 *G. hirsutum RING* genes exhibited differential expression in response to salt stress. The overexpression of the specific C3H2C3 *RING* gene, *GhZFRG1*, in Arabidopsis resulted in enhanced tolerance to salt stress. This study contributes to our understanding of the evolution of cotton RING ligases and paves the way for further functional analysis of the *RING* E3 ligase genes in cotton.

## 1. Introduction

The ubiquitin–proteasome system (UPS) represents a vital intracellular protein degradation system that is responsible for maintaining protein homeostasis and regulating a number of cellular processes [1]. The UPS is an enzymatic cascade reaction involving three crucial enzymes: ubiquitin-activating enzymes (E1s), ubiquitin-conjugating enzymes (E2s), and ubiquitin protein ligase (E3s). Among these, E3 ubiquitin ligases regulate the final step of the enzymatic cascade of ubiquitination and play a pivotal role in determining the specificity of target proteins for degradation [2]. In general, E3 ubiquitin ligases can be classified into four main types, namely HECT (homologous to E6-associated protein C-terminus), RING (Really Interesting New Gene), U-box, and CRLs (Cullin-RING ligases), based on their different structures and functions [1,3]. Of these E3 ligases, RING-type ligases have attracted particular attention due to their large number and diverse functional activities [3]. These ligases can be classified into two main categories: RING-HC (C3HC4) and RING-H2 (C3H2C3), along with five minor categories: RING-v, RING-C2, RING-D, RING-S/T, and RING-G [4].

Substantial evidence indicates that RING E3 ubiquitin ligases play an important role in various biological processes in plants [5]. A number of RING E3 ubiquitin ligases have been identified as key regulators of plant growth and development [6,7]. For example, in foxtail millet (*Setaria italica*), the C3HC4 RING E3 ligase SGD1, in conjunction with its E2 partner SiUBC32, has been demonstrated to ubiquitinate and stabilize the brassinosteroid (BR) receptor BRI1, thereby promoting plant growth and grain yield [6]. Similarly, the wheat *ZnF* (*TraesCS4B02G042900*) gene, which is orthologous to foxtail millet *SGD1*, has been demonstrated to have a conserved function in regulating grain yield via positive modulation of BR signaling [7]. In recent years, the roles of RING E3 ubiquitin ligases in plant tolerance to biotic and abiotic stress have also been extensively investigated [3,8]. In Arabidopsis, the C3H2C3 *RING* E3 ligase gene *SDIR1* is induced by drought and salt stress. The overexpression of *SDIR1* in Arabidopsis has been demonstrated to decrease tolerance to salt stress, but to enhance tolerance to drought stress by regulating the expression of stress-responsive genes involved in ABA signaling [9]. The C3HC4 RING ligase LbRZF1 from Limonium bicolor has been found to act as a positive regulator of salt gland development and salt tolerance, via modulation of the stability of LbCAT2 and LbMYB113 [10]. The C3HC4 RING ligase TaPIR1 from wheat has been found to act as a negative regulator of immunity by ubiquitinating and degrading TaHRP1, thereby suppressing chloroplast function and photosynthesis [11].

Cotton is the most important natural fiber crop, and is also cultivated as a food crop due to the high levels of vegetable oil and protein present in cottonseeds. However, high salinity can significantly impair the productivity and quality of cotton worldwide, particularly in China, where the predominant cotton-growing region is characterized by elevated salinity levels [12]. Previous research has indicated that RING E3 ubiquitin ligases are crucial in cotton’s response to salt stress. To illustrate, the *GhSARP1* gene, which encodes a C3H2C3 RING E3 ligase, demonstrated a reduction in expression in response to salt stress. Arabidopsis plants that overexpress *GhSARP1* exhibited reduced salt tolerance [12]. Furthermore, 140 RING-H2-type E3 ligase genes were identified in upland cotton, with some of them exhibiting responsiveness to salt stress [13]. However, further characterization of additional *RING* E3 ligase genes is required.

In the present study, we present the genome-wide identification and characterization of the RING E3 ubiquitin ligase family in *Gossypium hirsutum*, *G. arboretum* and *G. raimondii*. Furthermore, the gene structure, chromosomal locations and gene duplication analysis of each cotton *RING* E3 ligase gene were conducted. A total of 514, 509 and 914 *RING* genes were identified in *G. arboreum*, *G. raimondii* and *G. hirsutum*, respectively. It is noteworthy that 393 (|log_2_^FC^| ≥ 1 and *P*-adj ≤ 0.05) *G. hirsutum RING* genes were identified as being differentially expressed in response to salt stress based on transcriptomic data. Furthermore, a C3H2C3 *RING* gene, designated *GhZFRG1*, was confirmed to play a role in salt stress tolerance by overexpression in Arabidopsis. Therefore, this study provides useful insights for further investigation of the *RING* genes involved in salt stress response in cotton and other plant species.

## 2. Results

### 2.1. Identification of RING Domain Proteins in Gossypium Species

To identify proteins containing a cotton RING domain, two distinct methods were employed in this study. Initially, Blast searches were conducted against the protein sequences of the *Gossypium* species using the 509 Arabidopsis RING proteins as a query, with the default parameters. Subsequently, domain alignment analysis was conducted using the HMM profiles of RING domain proteins with *E*-values ≤ e^−10^. Consequently, a total of 927, 516, and 515 RING domains were identified in 914, 514, and 509 proteins in *Gossypium hirsutum*, *G. arboreum*, and *G. raimondii*, respectively. In *G. hirsutum*, 903 proteins were identified as containing a single RING domain, nine proteins exhibited two RING domains, and two proteins displayed three domains. In *G. arboreum*, 512 proteins were identified as containing only one RING domain, while two proteins were observed to possess two RING domains. In *G. raimondii*, 503 proteins were found to contain a single RING domain, while six proteins exhibited two RING domains. The comprehensive data regarding the Gossypium *RING* genes, including gene locus, chromosome location, exon number, sequence length, MW, and pI, are presented in Appendix A. The number of amino acids present in the RING domain proteins of *G. hirsutum* ranged from 102 (Ghir_D09G019080.1) to 4776 (Ghir_D12G014580.1). The MW of these proteins varied considerably, from 11.78 kDa (Ghir_A13G020890.1) to 534.99 kDa (Ghir_D12G014580.1). The pI of these proteins ranged from 3.56 (Ghir_D06G004860.1) to 10.68 (Ghir_A01G017410.1) (Appendix A).

Based on the amino acid residues at eight metal ligand (ml) positions and the distances between them, 927 *G. hirsutum* RING domain proteins were classified into seven RING types, namely RING-H2, RING-HCa/b, RING-v, RING-C2, RING-S/T, RING-D and RING-G (Table 1). Among these, RING-H2 and RING-HCa/b were the canonical RING domains with 551 domains (59.44%) and 288 domains (31.07%), respectively. This highlights the functional significance of these two types. In addition, RING-HC proteins were further subdivided into two subtypes, RING-HCa (276) and RING-HCb (12), according to the spacing patterns between ml7 and ml8. Furthermore, the remaining five modified RING types (RING-v, RING-C2, RING-S/T, RING-D and RING-G) account for only 9.49% of the total RING domains (Table 1).

### 2.2. Conserved Spaces Between Metal Ligand Residues in Cotton RING Domains

The eight ml residues of the RING domain are responsible for coordinating two zinc ions in a distinctive cross-brace structure. The ml1–ml2 and ml5–ml6 residues form a bond with one zinc ion, while the ml3–ml4 and ml7–ml8 residues bind with the other [14,15]. This distinctive configuration necessitates the conservation of specific regions within the ml pairs, including ml1–ml2, ml3–ml4, ml4–ml5, ml5–ml6, and ml7–ml8. In contrast, the regions between ml2-ml3 and ml6-ml7 exhibit greater variability. To assess the spacing patterns between different ml pairs, the number of amino acids between adjacent ml residues is carefully calculated. As illustrated in Figure 1, all 927 *G. hirsutum* RING domains have two, one, and two amino acids between ml1–ml2, ml3–ml4, and ml5–ml6, respectively (Table 1). Furthermore, 82.09% (761/927) and 98.27% (911/927) of the RING domains exhibit a two-amino-acid spacing between ml4 and ml5, and between ml7 and ml8, respectively (Figure 1A). In particular, all RING-v type domains contain seven amino acids between ml4 and ml5, whereas all RING-HCb type domains possess four amino acids between ml7 and ml8 (Figure 1A). Furthermore, the distance between ml2 and ml3 varied from 9 to 28 amino acid residues, with 15 (279/927), 14 (262/927), and 11 (222/927) being the most prevalent. The space between ml6 and ml7 comprised 6 to 45 residues, with 10 (533/927) being the most predominant spacing pattern (Figure 1B). These results are consistent with those previously reported in tomato and flax [4,16].

Notably, considerable variation was observed within the same RING type domains. As shown in Figure 1B, the RING-H2 domain contained the most variation, with 13 different spacing patterns, followed by the RING-HCa domain with eight different spacing patterns. Among them, 47.55% (262/551) and 42.29% (233/551) of the RING-H2 domains had 15 and 14 residues, respectively, while 74.64% (206/276) of the RING-HCa domains had 11 residues (Figure 1B). With respect to the ml6-ml7 pair, the RING-HCa domain contained 17 different spacing patterns, of which 10 (92/276), 6 (53/276), 11 (37/276) and 12 (26/276) residues were relatively common (Figure 1B). Specifically, the RING-v domain contained three spacing patterns (12, 13 and 15 residues) between ml6 and ml7, but approximately 90% (60/67) comprised 12 residues (Figure 1B).

### 2.3. Phylogentic and Gene Sturctural Analyses of the G. hirsutum RING Genes

To elucidate the evolutionary relationship of the *G. hirsutum RING* genes, the 927 RING domain sequences were aligned to construct a neighbor-joining (NJ) phylogenetic tree using the MEGA11 software [17]. As illustrated in Appendix A, all *RING* genes were primarily classified into two distinct clades, namely clade I (RING-H2 clade) and clade II (RING-HC clade) (Appendix A). This finding aligns with previous reports on Arabidopsis *RING* genes. It is noteworthy that RING-v domains were identified within the RING-H2 clade, while RING-C2, RING-G and RING-S/T domains were observed in the RING-HC clade. Notably, RING-D domains were present in both.

Prior research has demonstrated that exon–intron structural variations are a prevalent phenomenon in gene families, with many instances resulting in the emergence of paralogs with distinct functional characteristics [18]. Accordingly, a more detailed investigation into the structural characteristics of exon and intron composition of the coding regions of the *G. hirsutum RING* genes has been initiated. The number of exons in the *G. hirsutum RING* genes exhibited considerable variation, with a range of 1 (intronless) to 24 (Appendix A), which may be related to the diversification of their functions. In the case of *RING* genes with a RING-H2 domain, the most common exon count is one, with a frequency of 51.28% (Appendix A). Subsequently, exons of five and two were observed, with a frequency of 10.44% and 9.34%, respectively (Appendix A). *RING* genes with a RING-HCa domain exhibit the most exon and intron variation, with 23 distinct exon and intron patterns. For instance, 40 *RING* genes have a single exon, while two genes possess as many as 24 exons. Furthermore, the *RING* genes with a RING-v domain contain seven distinct exon and intron patterns, with 25, 14 and 13 genes having seven, two and eight exons, respectively (Appendix A).

### 2.4. Genomic Localization, Gene Duplication and Synteny Analysis of Cotton RING Genes

As illustrated in Appendix A, 907 (99.23%) of the *G. hirsutum RING* genes were located on 26 specific chromosomes. Specifically, 457 *RING* genes were found in the A-subgenome, while 450 genes were distributed in the D-subgenome. Furthermore, five genes were anchored on five A-subgenome scaffolds, and two genes were distributed across two scaffolds. It is noteworthy that the distribution of *G. hirsutum RING* genes on each chromosome was not uniform. Chromosome A05 had the highest number of *RING* genes, with 51, while chromosome A04 had the lowest number, with only 16.

To elucidate the underlying mechanisms governing cotton *RING* gene expansion, a gene duplication analysis was conducted using the MCScanX program [19] and the coding sequences of the *G. hirsutum RING* genes. In detail, 949 pairs of cotton *RING* genes were identified as having arisen through segmental duplication, while 24 pairs were identified as having originated through tandem duplication (Figure 2), indicating that segmental duplication may have played a more significant role in the expansion of the *G. hirsutum RING* genes than tandem duplication. Moreover, the Ka/Ks ratio was calculated for each gene pair, with the majority (98.87%) exhibiting a ratio less than 1, ranging from 0.028 to 0.986 (Appendix A). This finding indicates that the vast majority of cotton *RING* genes were subjected to purifying selection during the long evolutionary process. Furthermore, seven gene pairs (Ka/Ks > 1) may evolve under strong positive selection after duplication.

To investigate the synteny of *RING* genes, a collinearity analysis was performed between *G. hirsutum* and the other three plant species, including *A. thaliana*, *G. arboretum* and *G. raimondii*, employing the MCScanX tool [19]. In the present study, 914, 509 and 514 *RING* genes were identified in *G. hirsutum*, *G. raimondii* and *G. arboretum*, respectively (Appendix A). Furthermore, 508 *RING* genes had been previously identified in the *A. thaliana* genome [15]. In total, 2445 *RING* genes were employed to elucidate the synteny relationship in this study. The results demonstrated that there were 681 collinear pairs between *G. hirsutum* and *A. thaliana*, 798 pairs between *G. hirsutum* and *G. arboretum*, and 811 pairs between *G. hirsutum* and *G. raimondii* (Figure 3). It is noteworthy that 391 *G. hirsutum RING* genes are collinear with *RING* genes from the other three species (Appendix A).

### 2.5. Differential Expression Pattern of G. hirsutum RING Genes in Response to Salinity Stress

In order to identify salt-responsive *RING* genes, publicly available transcriptomic data from roots of the upland cotton genotype GX100-2 after different periods of salt treatment were employed to determine differential expression patterns under conditions of salinity stress [20]. Consequently, 842 (92.12%) *RING* genes were identified as being expressed in the GX100-2 roots after varying periods of salt stress (Appendix A). A total of 393 *RING* genes, including 107, 153, 273 and 90 *RING* genes at 1, 3, 12 and 48 h of salt stress, respectively, demonstrated differential expression patterns based on the cutoff of |log_2_^FC^| ≥ 1 and *P*-adj ≤ 0.05 (Appendix A). Furthermore, 144 *RING* genes were identified as being differentially expressed in a minimum of two salt stress time points (Figure 4). Of particular note are 13 *RING* genes, comprising 10 up-regulated and three down-regulated genes, which exhibited differential expression at the aforementioned four time points (Appendix A). In addition, 45 pairs of duplications were observed among the 144 *RING* genes, including 41 segmental duplications and 4 tandem duplications. Notably, each pair of duplicated *RING* genes shows the same expression pattern, although the expression levels are different (Figure 4). For example, the tandem duplication genes *Ghir_A13G020880* and *Ghir_A13G020890* are all up-regulated by salt stress, suggesting that these genes may have undergone functional redundancy through the process of tandem duplication (Figure 4).

### 2.6. GhZFRG1 Is Responsive to Salt Stress in Cotton

In a previous study, we identified a NAC transcription factor, designated *GhSNAC2* (GenBank accession No. KU759895, Ghir_A05G039480 in this study), whose expression level is significantly up-regulated under salt stress (Appendix A). Ghir_A08G025550 was found to interact with GhSNAC2 based on yeast two hybrid (Y2H) analysis (Appendix A). The coding sequence of *Ghir_A08G025550* comprises 420 nucleotides and encodes a protein of 139 amino acids. This protein features a conserved zf-RING_2 (PF13639) domain at amino acids 85–130. Consequently, we designated Ghir_A08G025550 as GhZFRG1 (*G. hirsutum* zf-RING_2 gene 1). Further analysis reveals the presence of a RING-H2-type standard motif (C-X_2_-C-X_15_-C-X_1_-H-X_2_-H-X_2_-C-X_12_-C-X_2_-C), indicating that GhZFRG1 is a typical RING-H2-type protein (Figure 5A). The subcellular localization of GhZFRG1 was predicted using the Plant-mPLoc software (http://www.csbio.sjtu.edu.cn/bioinf/plant-multi/, (accessed on 22 November 2023)) [21] and it was found that GhZFRG1 might be a nuclear protein. Subsequently, the subcellular localization of GhZFRG1 was determined by transiently expressing a GhZFRG1-GFP fusion construct in Arabidopsis protoplasts. The results demonstrated that green fluorescence from the GhZFRG1-GFP fusion was exclusively detected in the nucleus, confirming that GhZFRG1 is a nuclear protein (Figure 5B). Furthermore, the expression pattern of *GhZFRG1* was investigated in the upland cotton cultivar Lumian 451 under salt stress conditions. It was observed that the expression level of *GhZFRG1* increased significantly in response to salt stress, reaching a peak at 3 h after salt treatment and subsequently declining, although remaining at a relatively high level at 48 h (Figure 5C).

To investigate the function of the *GhZFRG1* in response to salt stress, we ectopically expressed *GhZFRG1* in Arabidopsis using the CaMV 35 S promoter. Transgenic lines OE2, OE7 and OE9 were selected for the salt assay on the basis of their high levels of *GhZFRG1* transcript, as determined by qPCR analysis (Appendix A). When Arabidopsis seedlings grown on Murashige and Skoog (MS) medium, no significant differences were observed between the WT and OE lines (Figure 6A). Nevertheless, when these seedlings were cultivated on MS medium supplemented with either 100 or 150 mmol/L NaCl, the growth of the OE plants was significantly superior to that of the WT plants, although all seedlings exhibited reduced growth rates (Figure 6A). In comparison to the WT plants, the root length per plant of the OE lines was significantly longer than that of the WT, with the exception of OE9 plants under the condition of 150 mmol/L NaCl (Figure 6B). Similarly, the fresh weight per plant of the OE lines was greater than that of the WT (Figure 6C). These results indicate that *GhZFRG1* can enhance the salt tolerance of Arabidopsis.

## 3. Discussion

### 3.1. Characterization of Cotton RING Genes

The RING E3 ubiquitin ligases, which act as either positive or negative regulators of protein degradation, have been the subject of extensive investigation in a number of plant species. These include 509 RING E3 ligase proteins in *Arabidopsis thaliana* [15], 574 in *Linum usitatissimum* [4], 469 in *Solanum lycopersicum* [16], 338 in *Prunus persica* [22], 715 in *Brassica rapa* [23], and 1255 in *Triticum aestivum* [24]. Nevertheless, a genome-wide analysis of *RING* E3 ubiquitin ligase genes in cotton has yet to be conducted. Upland cotton (*G. hirsutum* L.) represents the largest source of renewable textile fiber, accounting for over 90% of global production. The tetraploid *G. hirsutum* evolved from an occasional hybridization process between a diploid A-genome species, such as *G. arboreum*, and a diploid D-genome species, like *G. raimondii* [25]. In this study, we conducted a comprehensive analysis of the *RING* E3 ubiquitin ligase gene family in cotton to ascertain their potential roles in response to salt stress. Our findings revealed the existence of 914 *RING* E3 ligase genes in the *G. hirsutum* genome, representing an approximate 1.8-fold increase compared to the 514 genes identified in *G. arboreum* and 509 genes in *G. raimondii* (Appendix A). Furthermore, the *RING* E3 ligase genes represent approximately 1.30% of the predicted protein-coding genes in *G. hirsutum*, which is comparable to the proportions observed in *G. arboreum* (1.25%), *G. raimondii* (1.36%), *Solanum lycopersicum* (1.31%) [16], *Triticum aestivum* (1.26%) [24] and *Linum usitatissimum* (1.32%) [4], and slightly lower than that in *Arabidopsis thaliana* (1.85%) [15]. These findings indicate that the proportion of *RING* E3 ligase genes remains relatively stable despite the considerable variation in genome size and complexity observed across different plant species. Furthermore, the proportional stability of the *RING* E3 ligase genes across different plant species suggests that these genes may play a conserved role in plant biology.

The *RING* E3 ubiquitin ligase genes of upland cotton were classified into seven distinct types, namely RING-H2 (551), RING-HC (288), RING-v (67), RING-C2 (12), RING-S/T (6), RING-D (2) and RING-G (1). Of these, the RING-H2 and RING-HC were the most prevalent types, collectively accounting for 90.51%. These findings are in accordance with the results observed in Arabidopsis [15] and flax [4]. Some plant species have been reported to lack certain types of RING E3 ligases. For example, wheat RING E3 ligases were divided into four major types, with the absence of RING-C2, RING-S/T, and RING-D [24]. Conversely, additional types of RING E3 ligases were observed in some plant species. For instance, apple contains nine types of RING E3 ligases, including RING-mH2 and RING-mHC, which are specific to the apple RING E3 ligases [26].

Exon–intron structural variation has been identified as a pivotal factor in the evolutionary history of gene families [18]. These variations are generated by insertion and deletion events and are of great value for elucidating the evolutionary mechanisms that shape different gene families [13]. The exon–intron structure of the *G. hirsutum RING* E3 ubiquitin ligase genes displayed notable variation, with a range of 1 to 24 exons. The most prevalent exon numbers were one (325), two (94), five (81), three (79) and seven (76), collectively representing 71.66% of the total. Furthermore, a previous study reported that approximately 69% (96/140) of RING-H2 ligase genes lacked introns [13]. In the present study, we found that 50.82% (280/551) of RING-H2 E3 ligase genes also lacked introns. This discrepancy in the percentage of intronless genes may be attributed to the greater number of RING-H2 ligase genes identified in our study.

### 3.2. Duplication and Synteny Analysis of Cotton RING Genes

Gene duplication, encompassing both tandem and segmental duplication, represents the primary mechanism responsible for the expansion of specific gene families within plant genomes [16]. A gene duplication analysis of the *G. hirsutum RING* E3 ubiquitin ligase genes revealed the existence of 973 duplicated *RING* gene pairs, comprising 949 segmental duplication pairs and 24 tandem duplication pairs, in upland cotton (Figure 2). It can therefore be concluded that segmental duplication is the dominant mechanism responsible for the expansion of the *RING* E3 ubiquitin ligase genes in the *G. hirsutum* genome. These findings are in accordance with the results reported in the *Linum usitatissimum RING* genes [4]. Moreover, the overwhelming majority (966/973) of duplicated *RING* gene pairs exhibited evidence of strong purifying selection during evolution (Appendix A), suggesting that this process was a pivotal determinant in the restriction of the *G. hirsutum RING* E3 ubiquitin ligase genes. The synteny analysis of *RING* E3 ubiquitin ligase genes among *G. hirsutum*, *A. thaliana*, *G. arboreum*, and *G. raimondii* revealed that a greater number of orthologs were present between *G. hirsutum* and *G. raimondii* than between the other two plant species (Figure 3). It has been previously demonstrated that orthologous genes typically have equivalent functions in different organisms [27]. Our findings indicate that approximately 43% (391/914) of *G. hirsutum RING* genes are collinear with *RING* genes from the other three species, suggesting that these orthologs may possess analogous functions.

### 3.3. RING Genes Are Responsive to Salt Stress in Cotton

The detrimental effects of salt stress on the growth, yield and quality of cotton are well documented [28]. The objective of this study was to ascertain whether *RING* genes play a role in cotton’s response to salt stress. To this end, an analysis of the expression patterns of *RING* E3 ubiquitin ligase genes based on transcriptomic data from cotton roots after varying periods of salt stress was conducted. Our findings demonstrated that approximately 92% (842) of the 914 identified *RING* ligase genes were expressed in cotton roots (Appendix A). In addition, a set of 393 *RING* genes exhibited differential expression at various stages of salt treatment. Notably, 10 *RING* genes were found to be up-regulated, and three to be down-regulated, throughout the duration of the salt stress treatment, respectively (Figure 4), indicating that these *RING* genes may play a role in the salinity response in cotton.

Furthermore, a number of *RING* E3 ubiquitin ligase genes have been functionally validated and demonstrated to play a crucial role in fiber development and in response to environmental factors. *GhRING1* (*Ghir_A04G011720* in this study), the first *RING* E3 ligase gene identified in cotton, has been demonstrated to exhibit high expression levels in elongating fiber cells [29]. However, our findings indicate that *GhRING1* does not exhibit a response to salt stress (Appendix A). *GhSARP1* (*Ghir_A03G005310* in this study), which encodes a RING-H2 E3 ligase, has been reported to act as a negative regulator of cotton in response to salt stress [12]. This study revealed that *GhSARP1* was also down-regulated by salt stress (Appendix A). The RING-H2 ligase GhDIRP1 (Ghir_D03G013450 in this study) has been identified as a negative regulator involved in cotton resistance to *Verticillium dahliae* [30]. The results of this study demonstrated that *GhDIRP1* was induced by salt stress (Figure 4). These findings suggest that these cotton *RING* genes may play a role in the response to both biotic and abiotic stress. Moreover, *GhZFRG1* (*Ghir_A08G025550*) was found to also be induced by salt stress based on transcriptomic data (Appendix A), and this was subsequently confirmed by real-time quantitative PCR (qPCR) analysis (Figure 5C). The heterogeneous overexpression of *GhZFRG1* was observed to enhance the salt tolerance of transgenic Arabidopsis (Figure 6), which suggests that *GhZFRG1* may positively regulate the cotton plant’s response to salt stress. The disparate mechanisms observed between *GhZFRG1* and *GhSARP1* or *GhDIRP1* are likely attributable to their distinct structures, functions, and interactions with other proteins within cellular pathways [3]. Nevertheless, further investigation is necessary to confirm that GhZFRG1 is an active E3 ligase and to determine its precise role in abiotic stress tolerance.

## 4. Materials and Methods

### 4.1. Identification of the RING Genes in Cotton

The protein sequences of *Gossypium arboreum* (CRI) and *G. raimondii* (JGI) were obtained from the CottonGen database (https://www.cottongen.org/, (accessed on 20 November 2023)) [31,32,33], and the protein sequence of *G. hirsutum* (HAU) was downloaded from the CottonMD database (https://yanglab.hzau.edu.cn/CottonMD, (accessed on 20 November 2023)) [34,35]. To identify putative *RING* genes in cotton, two methods were employed. First, the 508 RING proteins reported in Arabidopsis thaliana were downloaded and used to perform a BLASTP search on the protein sequences of the above three Gossypium species with default parameters [15]. In addition, the Hidden Markov Model (HMM) profiles of RING domains (PF00097, PF12906, PF13639, PF13923, PF13920, and PF15227) were downloaded from the InterPro database (http://www.ebi.ac.uk/interpro/, (accessed on 20 November 2023)) [36], which was queried to identify RING protein sequences in the above three cotton species using the HMMER program (v3.3.2, http://www.hmmer.org/, (accessed on 20 November 2023)), with *E*-value ≤ e^−^^10^. All identified protein sequences were further confirmed using the CDD (https://www.ncbi.nlm.nih.gov/cdd/, (accessed on 22 November 2023)) and SMART (http://smart.embl-heidelberg.de/, (accessed on 22 November 2023)) databases [37]. Furthermore, the online software ExPASy (https://web.expasy.org/compute_pi/, (accessed on 22 November 2023)) was used to predict the MW (molecular weight) and pI (theoretical isoelectric point) of cotton RING proteins.

### 4.2. Phylogenic and Structural Analyses of the Cotton RING Genes

The RING domain sequences of the cotton *RING* genes were used for phylogenetic analysis. A neighbor-joining (NJ) phylogenetic tree was constructed using MEGA 11 [17] with 1000 bootstrap replicates. The exon–intron structures of the coding regions of the cotton *RING* genes were retrieved from the GFF files of three Gossypium genomes by using the TBtools-II (v2.142) software [38].

### 4.3. Chromosomal Mapping, Gene Duplication and Synteny Analyses

The chromosomal location information for each *RING* gene was extracted from the GFF files. The MCScanX software was used to determine duplicated *RING* genes within the *G. hirsutum* genome and to identify the synteny between *RING* genes in *G. hirsutum* and those in the other plant species [19,39].

### 4.4. Expression Profiles of RING Genes in Response to Salt Stress

The transcriptome data of upland cotton were obtained from the Sequence Read Archive (SRP343057) at the National Center of Biotechnology Information (NCBI) website [20], where the cotton plants were subjected to salt stress. In summary, the seedlings at the three-leaf stage of *G. hirsutum* cultivars GX100-2 were treated with 150 mM NaCl for 0, 1, 3, 12 and 48 h [20]. Subsequently, the root samples were collected for RNA-seq analysis. The expression data of *G. hirsutum RING* genes from the roots under salt stress was retrieved. Transcript levels were calculated as fragments per kilobase of exon model per million mapped fragments (FPKM). The identification of differentially expressed genes was conducted using the DESeq2 R packages (1.16.1), with a cutoff of |log_2_^FC^| ≥ 1 and *P*-adj ≤ 0.05. The heat maps were visualized using the TBtools-II (v2.142) software [38]. Furthermore, the expression of *GhZFRG1* was analyzed by real-time quantitative PCR (qPCR). Total RNA was extracted from the leaves of the Lumian 451 cultivar treated with 200 mM NaCl. The qPCR was conducted in accordance with the method previously described by Chen et al. (2023) [40]. The *ubiquitin* gene (GenBank accession No. AY189972) was employed as the reference gene [40]. The qPCR assay included three technical replicates and three independent biological replicates. The relative expression level of the *GhZFRG1* was calculated based on the 2^−ΔΔCt^ method [41]. The qPCR primers are listed in Appendix A.

### 4.5. Transient Expression of Recombinant GhZFRG1-GFP Protein in Arabidopsis Protoplasts

The 420 bp (coding sequence) of *GhZFRG1* was synthesized and subsequently incorporated into a pC1300s vector with the CaMV 35S promoter and enhanced green fluorescent protein (eGFP). The *pC1300s-35S:GhZFRG1-GFP* and *pC1300s-35S:Ghd7-RFP* (plant nuclear marker) constructs were introduced into Arabidopsis mesophyll protoplasts. Subsequently, the resulting fluorescence from the fusion proteins was detected using an Olympus FV 1200 confocal laser scanning microscopy (Olympus, Tokyo, Japan) one day after transformation.

### 4.6. Overexpression of GhZFRG1 in Arabidopsis

The coding sequence of *GhZFRG1* was inserted into the expression vector pC3300s, which is governed by the CaMV 35S promoter. The resulting *pC3300s-35S:GhZFRG1* construct was transformed into *Agrobacterium tumefaciens* strain GV3101 for Agrobacterium-mediated infiltration of *A. thaliana* (Col-0). The transgenic T_0_ seedlings were screened on MS medium containing 100 mg L^−1^ BASTA. The surviving seedlings were planted to generate T_1_ seeds. Twelve T_2_ lines with a segregation of 3:1 were selected and confirmed by qPCR (Appendix A). The functional analysis of *GhZFRG1* in response to salt stress was conducted using three T3 transgenic lines and WT Arabidopsis. The seeds of the transgenic lines and the WT were germinated on MS medium. Subsequently, the one-week-old seedlings were transferred to MS medium containing different concentrations of NaCl (0, 100 or 150 mM NaCl). After a period of five days, the root length and fresh weight were measured.

### 4.7. Statistical Analysis

To analyze the differentially expressed *RING* genes in response to salt stress, fold changes of various gene expression analyses and the related statistical calculations of the two tested conditions were performed using the DESeq2 package (1.16.1). To ensure the reliability of the results, all qPCR and functional analyses of *GhZFRG1* were conducted at least three times.

## 5. Conclusions

This study performed a genome-wide analysis of the RING E3 ubiquitin ligases in three *Gossypium* species. A total of 514, 509, and 914 *RING* genes were identified in the genomes of *Gossypium arboretum*, *G. raimondii*, and *G. hirsutum*, respectively. The expansion of the *G. hirsutum RING* genes appears to be primarily driven by segmental duplication. Furthermore, the majority of the duplicated *RING* genes had been subjected to purifying selection during the evolutionary history of cotton. Additionally, the expression analysis of 393 *G. hirsutum RING* genes under salt stress revealed their potential involvement in cotton’s response to high salinity. The overexpression of *GhZFRG1* led to enhanced tolerance to salt stress in Arabidopsis. Our results will contribute to a deeper understanding of the evolutionary mechanisms and functional diversity of *RING* genes in cotton, and will facilitate the selection of candidate *RING* genes for further functional analysis and potential genetic improvements of salt tolerance in cotton and other plant species.

## Figures and Tables

**Figure 1 ijms-26-00359-f001:**
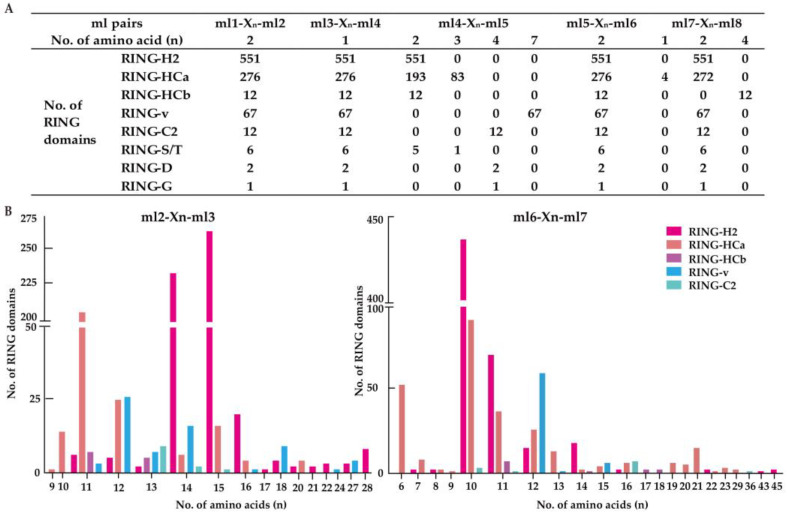
Distance variation between ml pairs in *G. hirsutum* RING domains. (**A**) Distance variation between ml pairs ml1–ml2, ml3–ml4, ml4–ml5, ml5–ml6, and ml7–ml8. (**B**) Comparison of the number of amino acids in the loops between ml2 and ml3, and ml6 and ml7 of the RING-H2, RING-HCa, RING-HCb, RING-v, and RING-C2 domains. ml denotes metal ligand. Xn indicates the number of amino acids observed between two conserved metal ligands.

**Figure 2 ijms-26-00359-f002:**
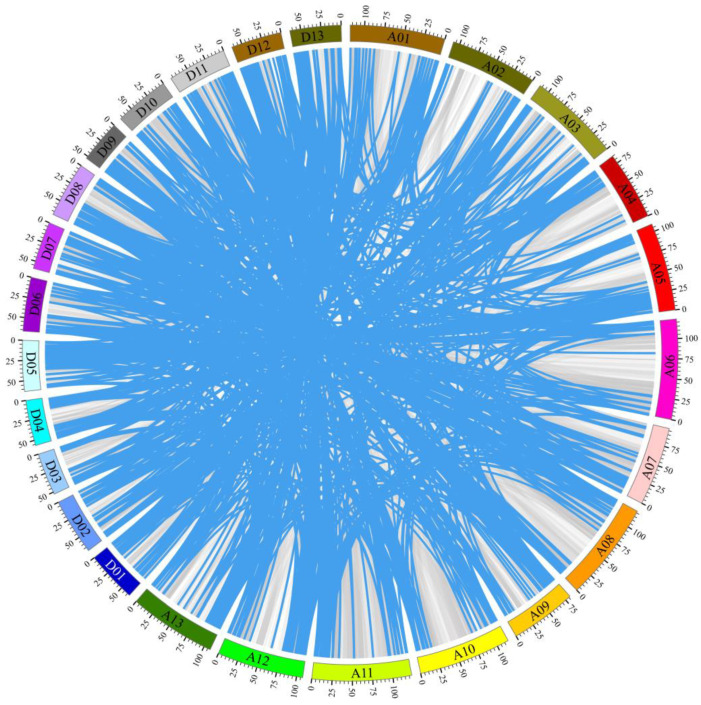
Duplicated *RING* gene pairs in the *G. hirsutum* genome. The 973 duplication pairs are indicated by blue lines (Appendix A). The chromosomes A01–A13 and D01–D13 are represented by different colors. The scale bar displayed on each chromosome denotes the chromosomal length in megabases (Mb).

**Figure 3 ijms-26-00359-f003:**
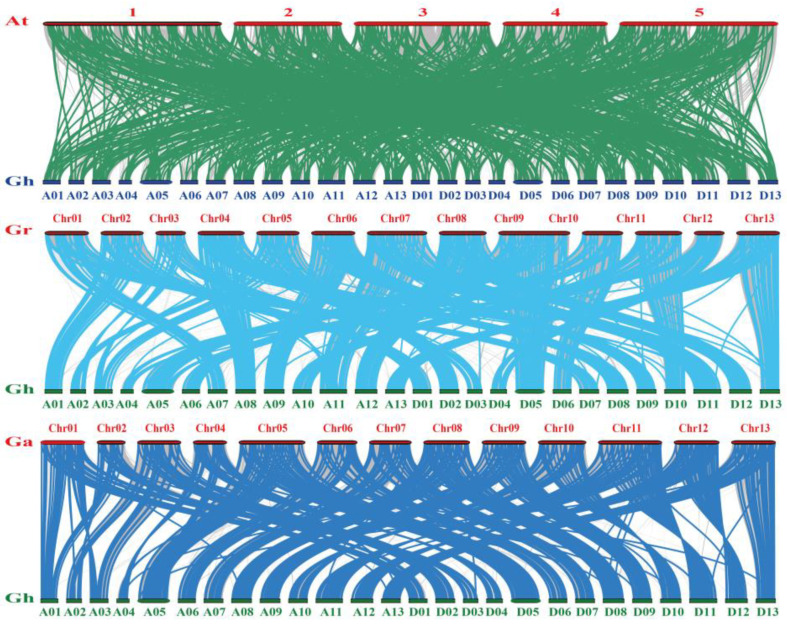
Synteny analysis of *RING* genes between *G. hirsutm* and other plant species. At, Gr, Ga, and Gh indicate *A. thaliana*, *G. raimondii*, *G. arboretum*, and *G. hirsutum*, respectively. The numbers 1 to 5, and 01 to 13 indicate different chromosomes. The grey lines in the background represent the collinear blocks within the respective compared genomes, while the colored lines show the collinear *RING* gene pairs (Appendix A).

**Figure 4 ijms-26-00359-f004:**
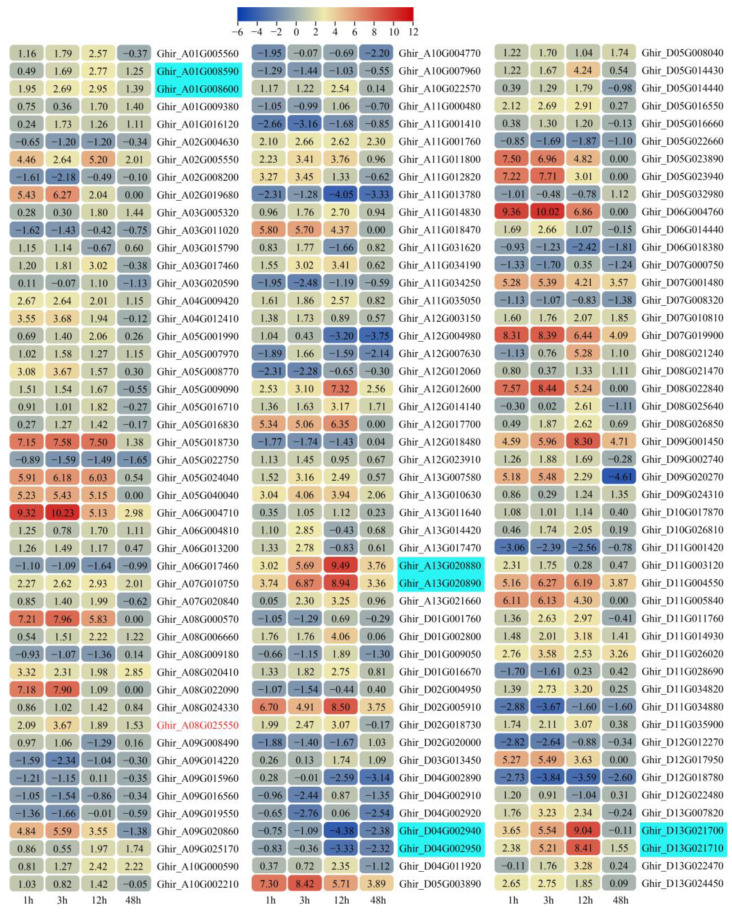
Expression profiles of the 144 *G. hirsutum RING* genes under salinity stress. Expression patterns of *G. hirsutum RING* genes determined from reanalysis of publicly available transcriptome data from the roots of cotton cultivar GX100-2 (SRP343057). The heat map was constructed based on transcript levels, calculated as fragments per kilobase of the exon model per million mapped fragments (FPKM). DEGs were identified using the DESeq2 R packages (1.16.1), with a cutoff of |log_2_^FC^| ≥ 1 and *P*-adj ≤ 0.05. The selection of these 144 *RING* genes was based on their differential expression in at least two salt stress time points. The values marked on round rectangles correspond to the log_2_^FC^ values. The scale represents the relative expression levels, with red indicating higher expression and blue indicating lower expression. The *RING* genes highlighted in pale blue are tandemly duplicated genes.

**Figure 5 ijms-26-00359-f005:**
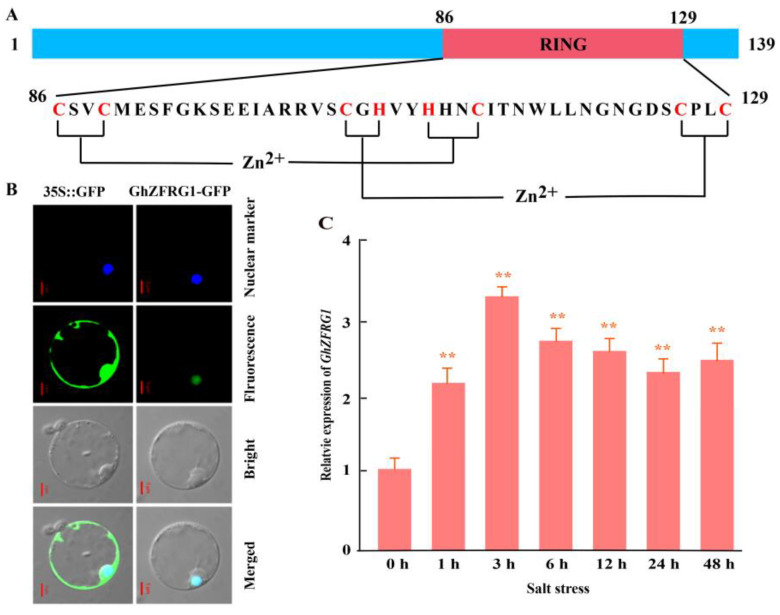
GhZFRG1 is a nuclear-localized RING-H2-type protein. (**A**) Diagram of the domain structure of GhZFRG1. The RING domain was highlighted in red. The amino acid sequence of the RING domain is presented below, with red amino acids indicating the conserved cysteine (**C**) and histidine (H) residues that form the zinc finger. (**B**) Subcellular localization of GhZFRG1 in Arabidopsis protoplasts. Transient expression of 35S::GhZFRG1-GFP or 35S::GFP was performed in Arabidopsis protoplasts. The scale bar is 5 μm. (**C**) Time course analysis of the relative GhZFRG1 expression levels in response to salt stress. The data represent the means ± SD of three independent assays. Significant differences were indicated as ** (*t*-test, *p* < 0.01) between the treatment and control.

**Figure 6 ijms-26-00359-f006:**
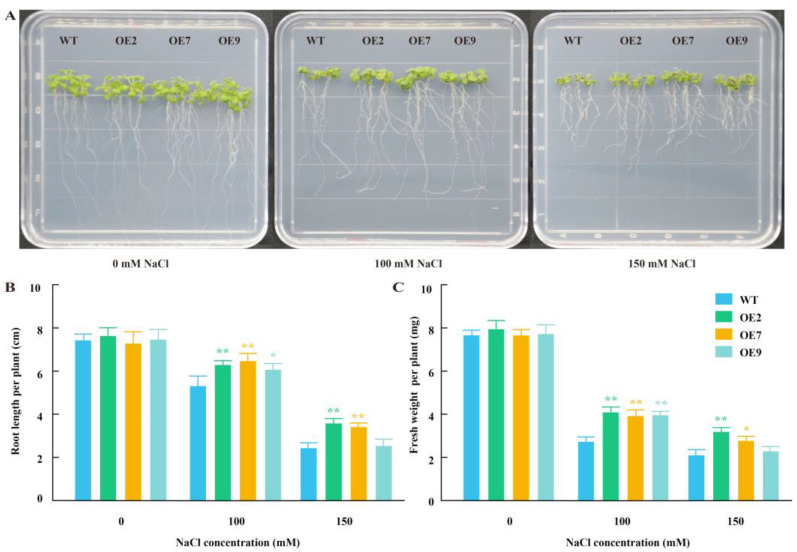
*GhZFRG1* enhances salt tolerance in Arabidopsis. (**A**) Seedlings from wild type (WT) and *GhZFRG1* overexpression (OE) lines grown under normal conditions (0 mM NaCl) or different salt stress treatments (100 or 150 mM NaCl). (**B**) The root length of the seedlings under the specified conditions. (**C**) The fresh weight of the seedling under the specified conditions. The three transgenic lines (OE2, OE7 and OE9) and the WT seeds were germinated on MS medium. Subsequently, the one-week-old seedlings were transferred to MS medium containing different concentrations of NaCl (0, 100 and 150 mM). Finally, the root length and fresh weight were measured after five days of growth. The data represent the means ± SD of three independent assays. Significant differences were indicated as * (*t*-test, *p* < 0.05) and ** (*p* < 0.01) between the WT and OE.

**Table 1 ijms-26-00359-t001:** The types and features of RING domains in *G. hirsutum*.

RING Domain	Consensus Sequence															
Type	No.	ml1		ml2		ml3		ml4		ml5		ml6		ml7		ml8
RING-H2	551	C	X_2_	C	X_11–28_	C	X_1_	H	X_2_	H	X_2_	C	X_7–45_	C	X_2_	C
RING-HCa	276	C	X_2_	C	X_9–20_	C	X_1_	H	X_2,3_	C	X_2_	C	X_6–29_	C	X_1,2_	C
RING-HCb	12	C	X_2_	C	X_11–13_	C	X_1_	H	X_2_	C	X_2_	C	X_11–18_	C	X_2_	C
RING-v	67	C	X_2_	C	X_11–27_	C	X_1_	C	X_7_	H	X_2_	C	X_12–15_	C	X_4_	C
RING-C2	12	C	X_2_	C	X_13–15_	C	X_1_	C	X_4_	C	X_2_	C	X_10–36_	C	X_2_	C
RING-S/T	6	C	X_2_	S/T	X_10–14_	C	X_1_	H	X_2,3_	C	X_2_	C/S	X_6,13_	C	X_2_	C
RING-D	2	C	X_2_	C	X_12_	C	X_1_	H	X_2_	C	X_2_	C	X_10_	C	X_2_	C
RING-G	1	C	X_2_	C	X_16_	C	X_1_	H	X_2_	G	X_2_	C	X_13_	C	X_2_	C

Note: The eight conserved metal ligand (mL) sites are presented as ml1 to ml8. The number of amino acids between the conserved ml residues is denoted by X_(n)_. The conserved cysteine and histidine residues are indicated by C and H, respectively.

## Data Availability

Data are contained within the article.

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
