# Peer review of "Systematic Analysis of Cotton RING E3 Ubiquitin Ligase Genes Reveals Their Potential Involvement in Salt Stress Tolerance"

_ijms, 2025, doi:10.3390/ijms26010359_

Round 1
Reviewer 1 Report
Comments and Suggestions for Authors
- Abstract (Lines 10-24): The abstract provides a good overview, but it could be more concise. Consider summarizing key findings in fewer words to enhance clarity and impact.
- Introduction (Lines 27-40): The introduction effectively sets the context; however, please ensure that the references are up-to-date. For instance, include recent studies from 2021 or later.
- Introduction (Line 62): The sentence regarding the importance of RING E3 ligases could benefit from a more explicit connection to cotton. Why is this particularly relevant for cotton compared to other crops?
- Results (Lines 79-90): The identification method for RING domain proteins is well-explained, but it would be helpful to include the specific criteria used for selection. This information would strengthen the methodology.
- Results (Line 106): The discussion of the RING types is informative; however, consider adding a brief explanation of the functional significance of each type. This would help readers understand their relevance.
- Results (Lines 165-174): The gene structure analysis is interesting, but the figures referenced are not clearly labeled. Ensure that all figures have descriptive captions for better comprehension.
- Discussion (Lines 272-292): The discussion section could be enhanced by linking findings back to the broader implications for cotton breeding strategies. How might these insights inform future research?
- Figure 1: The quality of Figure 1 could be improved. Ensure that all labels are legible and that the figure conveys information without requiring extensive reading of the text.
- Figure 5: Consider providing a legend that explains the significance of the colors used in the expression profiles, as this will aid in interpreting the data.
- Methodology: The methodology section lacks a detailed description of the statistical analyses used. Please provide more information about the statistical tests conducted to validate the findings.
- Conclusion: The conclusion could be more robust. Summarize the key findings and their implications for cotton research and potential applications in agriculture.
- References: Ensure that all references are formatted consistently according to the journal's guidelines. This includes checking for proper punctuation and citation style.
- Language and Clarity: Throughout the manuscript, there are several areas where the language could be simplified for better clarity. Consider revising complex sentences to enhance readability.
- Figures and Tables: Some tables, particularly Table S1, would benefit from clearer organization. Group related data together to facilitate easier comparison.
Future Research Directions: It would be beneficial to include a section on potential future research directions based on your findings. This could inspire further studies in the field of cotton genetics.
Reviewer 2 Report
Comments and Suggestions for Authors
In this manuscript, the authors identify RING E3 ubiquitin ligase genes in cottons comprehensively and describe their characteristics such as # of domain, exon numbers and duplications. They also provide a characterization of one of the RING family genes, GhZFRG1, including its salt-responsive expression in cotton, subcellular localization, and the ability to enhance salt tolerance when overexpressed in Arabidopsis thaliana. The overall organization of the manuscript is reasonable, but it also suffers from several major shortcomings. Some figures on gene characteristics are very descriptive and their significance are not clear. The molecular analysis of GhZFRG1 itself is OK, but the reason for focusing on GhZFRG1 out of all the salt-responsive candidate genes is not explained. Important details regarding some experiments are missing or need to be clarified. Below is a list of suggestions that need to be addressed before publication.
Specific comments
(1) Fig. 2: descriptive and the significance of exon counting in the context of this paper is not clear. If you pick any gene family, you will see a certain degree of exon number variation among them. Is anything special about exon number variation in the RING family genes? How about introns? (single exon vs intron-less genes?)
(2) With the high number of single exon genes, is RING-H2 enriched with potential pseudogenes?
(3) Fig. 3: What should a reader learn from this figure? L. 180-181 says that 949 pairs and 24 pairs are segmental duplication and tandem duplication, respectively. However, neither of the information can be obtained from looking at the figure. A similar issue applies to Fig. 4 – “the grey lines in the background” cannot be seen.
(4) Fig. 5: It is not clear how they performed the transcriptome analysis. Was it their own experiment, or did they use a previously published dataset? No methods or references are available. Please describe the source of the data, experimental procedures (i.e. library preparation, NGS platform, etc.) and bioinformatics (raw read processing to DGE analysis).
(5) From Section 2.6., it is unclear why the rest of the manuscript focuses on GhZFRG1 (Ghir_A08G025550). Why did you pick a gene that is less obvious in salt stress response? Wouldn’t Ghir_A09G001540 or Ghir_D09G001450 be better candidates? Please provide a reasonable justification that connects previous figures to Fig. 6-7.
(6) Fig. 7: the procedure needs to be explained further. Were they germinated on the NaCl containing media? If so, did the OE lines show any difference in germination rate under salt stress?
(7) The overexpression of GhZFRG1, a RING-H2 gene, is improving salt tolerance in Arabidopsis while another RING-H2 ligase GhSARP1 overexpression did the opposite (ref 12). The discussion also mentions another RING-H2 protein (GhDIRP1) as a negative regulator in biotic stress. To highlight the difference, it would be informative to discuss potential mechanisms in which GhZFRG1 acts as a positive regulator.
(8) It would be ideal to demonstrate that the GhZFRG1 is a bona fide E3 ligase, like done in the ref [12]. Could it be done in a reasonable amount of time? If not, the limitation should be mentioned in the discussion.
(9) l.342-343, l.356-357: These studies, including this one, collectively show a variety of expression patterns of RING family genes during salt stress. I don’t think this automatically means that “these RING genes may play a crucial role”. Please tone down.
Minor comments
l.66-67: this sentence is not clear. “(a gene) is diminished in response to salt stress.” needs to be clarified using more specific terms.
l.71: “is require.” should be “is required”.
l.115: “as illustrateds” should be “as illustrated”
l.134: Remove an extra space in “ml 6”
l.146: Remove an extra space before a comma in “distinct clades ,”
Fig. 5: Please add a label to the color scale bar.
l. 228: Please spell out what “ZFRG” stands for.
Fig. 6C: please clarify how the statistical significance was calculated.
Reviewer 3 Report
Comments and Suggestions for Authors
Dear authors,
The manuscript by Li et al documents the roles of the cotton RING E3 ubiquitin ligase genes under salt tolerance in cotton. The authors performed systematic identification, characterization, and expression analysis of RING genes in cotton. They found that segmental duplication may be the primary mechanism responsible for the expansion of the cotton RING E3 ligase gene family. Therefore, the authors analyzed expression of RING E3 ligase genes under salt stress. One of the genes, GhZFRG1 was confirmed to increase salt tolerance by overexpression in Arabidopsis. Finally, the authors concluded that the evolution of cotton RING E3 ubiquitin ligase genes could improve its salt tolerance in cotton.
The following comments are suggestions that are important to address to improve the overall quality of this manuscript.
Major comments:
1. Every result or conclusion should be indicated by the data resource.
2. To explain why Ghir_A08G025550 or GhZFRG1 has been selected to study in the end of result 2.5 or beginning of result 2.6
Minor comments:
Line 96-98. Add a and b after RING-HC to match with table 1.
Line 117-118. Add reference (table or fig.) after respectively to indicate where the data are from.
Line 121-122. As concern as that in line 117-118.
Line 129-132. As concern as that in line 117-118.
Line 157-158. As concern as that in line 117-118.
Line 195-196. As concern as that in line 117-118.
Line 267-268. Revise this sentence. For example, under normal condition (0 mM NaCl) or different salt stress treatments (100 mM, 150 mM NaCl).
Line 269. 1) to explain briefly seeds are planted directly or seedlings are transferred to the plate. 2). Fresh/dry weight instead of weight in panel C and legend.
Comments on the Quality of English LanguageThe writing should be revised by the native.
Reviewer 4 Report
Comments and Suggestions for Authors
Comments for authors
Title
- The title is clear and provides a concise summary of the study's scope and findings. However, it could be slightly refined to enhance readability and engagement. Such as “Systematic Analysis of Cotton RING E3 Ubiquitin Ligase Genes Reveals Key Mechanisms in Salt Stress Tolerance”.
Abstract
- While "RING" is introduced as "Really Interesting New Gene," this phrasing might detract from the scientific tone. Consider omitting "Really Interesting New Gene" and simply referring to it as "RING E3 ubiquitin ligases," as this is already an established term in the field.
Introduction
- Line 32: The passage transitions from general descriptions to specific details (e.g., from UPS to RING ligases) without clear signposting. Adding transition phrases could improve the flow, such as "Among these E3 ligases, the RING-type has garnered particular attention due to...".
- Throughout manuscript model plant name scientific names are non-italicise.
- There are minor grammatical and typographical error, such as "enchance toleracne" instead of "enhance tolerance" (line 54).
- In the last passage of introduction, the study's objectives are clearly stated, with a focus on genome-wide identification and characterization of the RING E3 ubiquitin ligase family in multiple cotton species, however, following important aspects are missing which should incorporated;
· Highlight the novelty of the study and its contributions to understanding cotton genetics and stress responses.
· Briefly mention key findings or insights derived from the genome-wide analysis.
· Specify the criteria or methods used in the transcriptomic analysis to identify salt-stress-related ligases. Specifying how these putative ligases were identified (e.g., expression patterns, stress-induced regulation) would provide more depth.
M&M
- The Materials and Methods section is missing a Statistical Analysis subsection, which raises concerns about the reliability and rigor of the study.
- Line 399: Minor grammatical error, such as "analyze" instead of "analyzed" need correction.
- Section 4.4: For qPCR analysis, specify the internal reference gene used.
- Section 4.6: The concentration of BASTA used (100 mg L^−1) is provided, but more details on the screening protocol (e.g., duration of selection, number of cycles) would help clarify the methodology.
- Section 4.6: While the transgenic lines are to be analyzed for salt stress response, the specific conditions (e.g., NaCl concentration, duration of stress treatment) are not mentioned. Providing these details would give readers a clearer understanding of the experimental setup.
- Section 4.6: It is mentioned that T3 homozygous lines are used, but the method for selecting these homozygous lines from the progeny is not described. Including the screening procedure for homozygous lines (e.g., PCR, phenotypic analysis) would add transparency.
Results
- Table 1 is unclear, as the description provided is insufficient for readers to fully understand its content, and the footnotes are missing to explain the meanings of "C" and "H."
- Line 124: This endorsing statement should be included in the discussion section.
- Line 174-177: These statements should be placed in discussion with proper references.
- Line 253-255: You could elaborate on how the transcript levels were quantified or validated to strengthen the description of the selection process.
- Line 260-262: It might be helpful to mention whether statistical tests were conducted to determine the significance of this difference.
Discussion
- The discussion section lacks specificity and does not highlight the study's novel contributions. It is recommended to enhance this section by emphasizing the unique aspects of the research and organizing the text under appropriate headings.
- Please also check similarity index of the manuscript.
Round 2
Reviewer 2 Report
Comments and Suggestions for Authors
The authors have improved the manuscript by clarifying some method details and potential limitations. However, it still has several issues which need to be addressed before publication.
The revised title “Reveals Key Mechanisms in Salt Stress Tolerance” is too generic and clearly an overstatement. This study does not provide any conclusion on the functionality of these E3 genes in cotton. Please make it a concise summary/conclusion of what this study supports without exaggeration – something like “Reveals their potential involvement in salt stress tolerance”.
Response to (1) and (2)
I understand the general concept, but I am not sure about your presentation. After the general statement in p.161-163, it goes on to describe exon number variation and that’s it. The functional diversity of the RING-domain containing genes has been well established in other studies/systems and is not novel or necessary to be supported by a descriptive figure like Fig. 2. What does the exon number distribution (Fig.2) mean for the GhRING gene family specifically? Does the GhRING gene family show higher variation than expected from a gene family of similar size? Do you find any correlation between specific exon structure with new functions in cotton? Currently I don’t find the answer to these in the manuscript and therefore am questioning its importance as a main figure. I suggest moving the figure to a supplemental figure.
> A RING gene with a single exon indicates an intronless RING gene
This statement needs further clarification. Intron can be in the 5'-UTR or 3'-UTR, without dividing an exon. Therefore, single exon does not always mean the gene is intronless. A gene can have multiple introns at UTRs with a single exon. That is why I requested distinction between single exon with UTR-introns vs intron-less genes as the latter have higher likelihood of being pseudogenes as you pointed. L.220 says 814 genes are expressed. Is intron-less genes enriched in the non-expressed genes (914-814=100 genes)?
Response to (3)
Please mention Table S2 and S3 in the corresponding figure legends to make it clear to readers that they have resources to look into. Also, the grey lines in Figs. 3/4 are nearly impossible to see when printed. Please change the color scheme so that they can be more visible.
Response to (4)
In section 4.4, please include details regarding RNA extraction and library preparation.
Per the journal’s guideline, the RNA-seq data should be deposited to a public database and its availability should be disclosed in the Data Availability Statement (l.503).
From IJMS instruction: "Data Availability Statement: In this section, please provide details regarding where data supporting reported results can be found, including links to publicly archived datasets analyzed or generated during the study.” Also, the Supplementary Materials, Data Deposit and Software Source Code says: “Deposition of Sequences and Expression Data; Manuscripts will not be published until the accession number is provided. New high throughput sequencing (HTS) datasets (RNA-seq, ChIP-Seq, degradome analysis, …) must be deposited either in the GEO database or in the NCBI’s Sequence Read Archive (SRA).”
Response to (5)
l.240: A citation to “previous study” should be clarified. Also, l.243 (unpublished data) is not acceptable. Please show the Y2H data to justify the focus on GhZFRG1. Can you verify the salt stress responsive expression of GhSNAC2 in your RNA-seq or by RT-qPCR? If so, it can be presented together to make the transition.
Response to Fig. 6C
In section 4.6, please clarify how you calculated the fold change for RT-qPCR analyses in Fig. 6C and S2. In Fig. S2, what do you mean by “relative expression” of GhZFRG1 compared to WT? I suppose WT is a non-transformed Arabidopsis thaliana. There should not be any GhZFRG1 expression to compare.
Reviewer 4 Report
Comments and Suggestions for Authors
No further comments.
Author Response
Comments 1: No further comments.
Response: We thank the reviewer for the positive comments.
Round 3
Reviewer 2 Report
Comments and Suggestions for Authors
The revision has addressed most of the major concerns, except one regarding the Data Availability Statement and RNA-seq data deposition. The IJMS policy is fair and clear. “Research is ongoing” is questionable as a justification because the manuscript contains the full analysis. Table S4 is your analysis, and the point of raw data deposition/availability is to allow readers to reanalyze the data themselves. If you claim that the study is ongoing and incomplete now, then the manuscript wouldn’t be ready for publication. Please read relevant sections of the IJMS instruction carefully and be compliant with the requirement, including the Data Availability Statement which cannot be “Not applicable (l.523)” in this case.
https://www.mdpi.com/journal/ijms/instructions#suppmaterials
· From IJMS instruction: "Data Availability Statement: In this section, please provide details regarding where data supporting reported results can be found, including links to publicly archived datasets analyzed or generated during the study.”
· Also, the Supplementary Materials, Data Deposit and Software Source Code says: “Deposition of Sequences and Expression Data; Manuscripts will not be published until the accession number is provided. New high throughput sequencing (HTS) datasets (RNA-seq, ChIP-Seq, degradome analysis, …) must be deposited either in the GEO database or in the NCBI’s Sequence Read Archive (SRA).”
>Response to (5)
Fig.S3B can be explained better. What do the red asterisk and all the numbers above each spot mean?
To make the transition better, I think it would be beneficial to mention the fold change behavior in the RNA-seq (Fig. 4, Table S4) and describe how the qPCR results (Fig. 5C) validate it, at around l.260.
>The revised Figure S4
I appreciate your correction. However, the new Fig.S4 shows incomplete information - for example, if OE1 expression were at WT level, the plot would mean that none of them are overexpressing. The 2-ΔΔCt with WT is OK as long as you describe clearly how you handled the zero or near-zero expression (large Ct values) of the WT in the calculation. Another supplemental table showing the Ct values for both Ubi and GhZFRG1 in WT and all analyzed transformants would be fine too.
Round 4
Reviewer 2 Report
Comments and Suggestions for Authors
The revised manuscript replaced their own RNA-seq experiment with re-analysis of a publicly available dataset, which showed results similar to theirs. As such, the data availability statement concern is no longer applicable. Please clarify in the Fig 4 legend that this is re-analysis of the previous study with the SRA-ID SRP343057.
Unfortunately, their response to the qPCR validation of OE lines made it more confusing than before. As the overexpression level of GhZFRG1 in the transgenic Arabidopsis lines is an important parameter in Fig. 6, it must be addressed appropriately before publication as outlined below.
>“If we calculate the ΔΔCt of the OE lines, we assume that the ΔCt (CtGhZFRG1-CtUbi) of the WT is zero.”
This part of their response and the corresponding revised text (l.504-506) do not make sense and should be corrected. To avoid any further confusion, I suggest authors provide a simple table of the Ct values for both GhZFRG1 and Ubi genes in WT and OE1~12 as Table S7 and remove Figure S4.
In the following 2-ΔΔCt Fold change calculation,
2-ΔΔCt Fold change = [2-(CtGhZFRG1-Ctubi)] of OE / [2-(CtGhZFRG1-Ctubi)] of WT
Treating the highlighted potion as zero makes no sense as it forces the denominator [2-(CtGhZFRG1-Ctubi)] of WT to be1.0 (2-0). The ΔCt (CtGhZFRG1-CtUbi) for WT being 12 is OK, as long as you clarify what it means (ie. the CtGhZFRG1 value for WT represents amplification from a background noise as there is no transgene in WT).
The clarification of Ct values as a table has additional values to readers. For example, readers can understand the GhZFRG1 qPCR background level (Ct=32 in WT). The value implies that there may be nonspecific amplification using the GhZFRG1 qPCR primers in the absence of the correct target. Because of this, [2-(CtGhZFRG1-CtUbi)] of WT will not be “near-zero” (this would happen when the CtGhZFRG1 value is very large like 40~45, or not available due to no amplification). The Ct values allow the interested readers to interpret the results on their own.
Round 5
Reviewer 2 Report
Comments and Suggestions for Authors
The revised Table S7 clarifies the underlying data and should be sufficient for the purpose.
Before publication, please clarify either at l.462 and/or l.497 that the WT background noise for GhZFRG1 was used in the calculation. A statement like below would suffice;
"To examine GhZFRG1 transgene expression relative to non-transgenic WT, the detected Ct value of GhZFRG1 in WT (32.12 on average), presumably a background noise, was used in the 2^(-ΔΔCt) method with normalization to the Ubi gene."
Author Response
Comments:
The revised Table S7 clarifies the underlying data and should be sufficient for the purpose.
Before publication, please clarify either at l.462 and/or l.497 that the WT background noise for GhZFRG1 was used in the calculation. A statement like below would suffice;
"To examine GhZFRG1 transgene expression relative to non-transgenic WT, the detected Ct value of GhZFRG1 in WT (32.12 on average), presumably a background noise, was used in the 2^(-ΔΔCt) method with normalization to the Ubi gene."
Response: Thank you for your helpful suggestion. A statement has been incorporated into lines 495 to 498 of the revised manuscript. Thank you again for your valuable suggestions to improve our manuscript.